# TEPEAK: A novel method for identifying and characterizing polymorphic transposable elements in non-model species populations

Devin Burke[1]*, Jishnu Raychaudhuri[1], Edward Chuong[1,2], William Taylor[2,3,4], Ryan Layer[1,5]

**1** BioFrontiers Institute, University of Colorado Boulder, Boulder, Colorado, United States of America, **2** Department of Molecular, Cellular, and Developmental Biology, University of Colorado Boulder, Boulder, Colorado, United States of America, **3** Department of Anthropology, University of Colorado Boulder, Boulder, Colorado, United States of America, **4** Museum of Natural History, University of Colorado Boulder, Boulder, Colorado, United States of America, **5** Department of Computer Science, University of Colorado Boulder, Boulder, Colorado, United States of America

\* devinburke0@gmail.com

## Abstract

Transposable elements (TEs) replicate within genomes and are an active source of genetic variability in many species. Their role in immunity and domestication underscores their biological significance. However, analyzing TEs, especially within lesser-studied and wild populations, poses considerable challenges. To address this, we introduce TEPEAK, a simple and efficient approach to identify and characterize TEs in populations without any prior sequence or loci information. In addition to processing user-submitted genomes, TEPEAK integrates with the National Center for Biotechnology Information (NCBI) Sequence Read Archive (SRA) to increase cohort sizes or incorporate proximate species. Our application of TEPEAK to 257 horse genomes spanning 11 groups reaffirmed established genetic histories and highlighted disruptions in crucial genes. Some identified TEs were also detectable in species closely related to horses. TEPEAK paves the way for comprehensive genetic variation analysis in traditionally understudied populations by simplifying TE studies. TEPEAK is open-source and freely available at https://github.com/ryanlayerlab/TEPEAK.

### Author summary

Transposable elements are DNA sequences that can move within genomes and generate genetic diversity. Despite their importance for evolution and adaptation, they remain poorly characterized in many species, especially those without extensive genomic resources. We developed TEPEAK, an open-source framework that identifies and analyzes transposable elements across populations without

**Data availability statement:** Gene and TE annotation data - 10.6084/m9.figshare.30569975 In order to fully replicate the experiment described in the manuscript TEPEAK only requires the list of SRA numbers listed in S2 Table. This data is publicly hosted on NCBI SRA and TEPEAK will automatically download the given data files and process them.

**Funding:** The author(s) received no specific funding for this work.

**Competing interests:** The authors have declared no competing interests exist.

needing prior knowledge of their sequences. TEPEAK detects insertion patterns shared among individuals, enabling large-scale comparisons of genome variation and its functional consequences.

We applied TEPEAK to 257 horse genomes spanning 11 populations to showcase its capabilities. Our results revealed clear population structure based on element insertions and showed that domestication and selective breeding have altered the diversity and frequency of these elements across breeds. We identified insertions overlapping genes related to behavior, muscle function, and neurological traits—features often targeted during domestication. By extending our analysis to closely related equids, we also traced the evolutionary persistence of several element families. TEPEAK provides an accessible approach for exploring how mobile DNA shapes population diversity, adaptation, and the genomic legacy of domestication.

## Introduction

While there is a growing appreciation for the impact structural variation (SV) has on genetic and phenotypic diversity, accurately identifying the many types of structural variation remains challenging [1,2]. Among SVs, transposable elements (TEs) are no exception. TEs are mobile genetic elements which can independently insert into alternate genomic regions, thereby multiplying throughout the host genome and occupying a substantial portion of many species' genomes [3]. Despite their significance, and the availability of a large number of sequenced samples, TEs are poorly cataloged in many non-model species, including those that remain a source of SVs [1]. This gap is due in part to the lack of simple and efficient methods available to identify and accurately differentiate polymorphic and novel TE insertions.

TEs are observed in nearly every eukaryotic genome and have profound implications due to the unique biological characteristics and the complementary relationship between environmental and population genetic factors. For example, insertions of TE sequences into coding regions can cause expansion and propagation of novel genes [4,5] and drive adaptive phenotypic variation from environmental pressures like global warming, disease and the adaptive immune system, speciation, and inbreeding and endangered populations [4,6,7]. Many types of TEs demonstrate locational specificity, with some targeting regions that minimize deleterious effects and others maximizing their propagation [4,8]. These behaviors are likely due to the evolutionary stability of TEs [9]. While TE sequences decay over time, often resulting in the loss of functionality, they often retain promoters and other remnant genomic features that indirectly impact nearby genes [8], including rewiring entire gene networks [5].

TEs exhibit remarkable diversity, with over 4 million different families in the most recent Dfam (3.9) database [10,11]. These families are distinguished into two different major classes; class I - retrotransposons and class II – DNA transposons. Class I elements involve an RNA intermediate, which undergoes reverse transcription

yielding a DNA copy that is inserted into a new location [5,8,12]. Most Class II DNA transposons mobilize via a "cut-and-paste "mechanism by a transposase, whereas rolling-circle Helitrons are an exception and transpose through a "peel-and-paste" process that does not use a transposase [12,13]. These two major classes are further divided into superfamilies, families, and subfamilies/subclasses [5,8,14].

Traditionally, TE families are defined as groups of homologous sequences that are presumed to derive from a common ancestral element. In practice, bioinformatic curation efforts have operationalized this concept using the so-called "80-80-80 rule," which classifies sequences sharing ≥80% mutual coverage and ≥80% identity as belonging to the same family. While this rule provides a pragmatic sequence-based heuristic for automated classification, it does not always capture the true evolutionary relationships among elements, particularly in highly divergent or recently expanded families [3,11]. Subclasses are then classified by mechanisms of replication or integration. They can be broadly distinguished by whether the element encodes for enzymes which facilitate self-transposition [3,5,12]. However, this classification does not encapsulate TE families that have emerged 'de novo' as is the case with short interspersed nuclear elements (SINEs) that are often derived from noncoding RNAs. Many of these SINE elements in fact hijack long interspersed nuclear elements (LINEs), which do encode proteins for autonomous transposition [8]. It has been shown that SINEs often share homology with LINEs, thereby explaining the capacity for LINEs to reverse transcribe and integrate SINEs [8].

TE discovery can roughly be divided into identification of polymorphic insertions of previously annotated TE families and insertion of novel unannotated TE families [6]. Current polymorphic insertion methods use both sequence homology-based and 'de-novo' structural feature searches to identify polymorphic insertions from a sequence or structural motif of interest [6,15–20]. However, these methods struggle to identify TEs different from those previously identified [21]. Although polymorphic TE families typically represent more recent insertions and are thus less degraded than fixed copies, sequence truncation and local decay still occur to varying degrees. Such heterogeneity complicates comprehensive extraction of all polymorphic insertions from short-read data, as partially decayed or fragmented elements may escape detection or be misclassified [3]. Using expert domain knowledge or 'de novo' methods is usually required to identify novel unannotated TE families [21]. This bottleneck results in a small and isolated perspective of the TE landscape in a species [22].

Another consequence of TE sequence decay is the prevention of TE annotation processes from entirely avoiding manual curation [6]. Both homology-based and 'de-novo' based structural feature searches struggle to identify TEs distinct from those already well established and described [9,11]. Current methods also often require expert knowledge of the TE landscape in the species of interest [21,23–25]. Researchers can utilize a sequence or transposon structural motif to extract polymorphic TE insertions in samples, but this yields only a small isolated perspective of the TE landscape and often fails to discover heavily mutated sequences and novel TEs altogether [6,11,21].

Extracting polymorphic insertions remains a critical component of constructing the entire TE landscape for a species. However, the bottleneck of current methods lies in their resulting narrow perspective. Most current methods require the user to provide a list of one or several TE family consensus sequences to detect polymorphic insertions from that family, though recent tools have been developed to circumvent this limitation by using pangenome structural variants to build TE libraries de novo [17,26]. Focusing on only known TE targets in a single species prevents a comprehensive and unbiased analysis of TE evolutionary relationships [11]. In addition to the high frequency of TE involved horizontal transfers, TE families have a complex evolutionary history. This means that single species studies are often insufficient to fully understand a specific TE's significance [22,27–35]. Without comparing closely related species, information pertinent to a TE's evolution, frequency, impact, and even fixation in a species cannot be obtained [36,37]. Recent advances have allowed for a drastic increase in the rate of new reference assembly releases, which has created a dilemma as the number of releases far outpaces annotation efforts [11]. This means that TE landscapes of non-model organisms are poorly characterized.

Even with the emergence of more TE focused population genetic studies, the lack of a streamlined process to effectively compare TEs in a population makes it difficult to generalize findings or even carry perspectives learned to other

studies [22,28,30–33,38,39]. There is a need for a method that yields the entire landscape of polymorphic TEs in a species. Such a method would have the potential to not only discover novel TEs, but also highly polymorphic TEs that may not be apparent with the limited focus of other methods. This method should also seamlessly allow comparison with closely related species to augment a near complete understanding of the significance of all TEs in the landscape. Together this method would allow for phylogenetic analysis of multiple TEs across multiple species.

We present TEPEAK, a simple streamlined process that is able to identify polymorphic insertions of annotated TEs and novel TEs. TEPEAK requires only a chromosome-level reference assembly—no prior knowledge of the TEs of interest is needed—making it well suited for discovering both known and novel transposable elements in non-model organisms. The process is based on detecting overrepresentation of SVs of specific lengths, which are likely to correspond to polymorphisms derived from TEs. TEPEAK uses a peak finding algorithm to identify clusters of TE families showing evidence of polymorphism from structural variant insertion data without excluding low copy number TEs. We then use these clusters of TE families to identify all relevant loci in the population. The output of our process is a simple file containing shared loci among the population as well as polymorphic sequences between samples at the same loci. These clusters of TE families can then be used to compare with any other species input into the pipeline.

We demonstrate TEPEAK's utility by reporting on the TE landscape of 257 horse genomes. We also show that our method identifies polymorphic insertions of the same Equid TE families in black rhino, wild ass, and plains zebra, several of which have not been previously annotated. Finally, we demonstrate our method is able to identify novel TEs in non-model species like black rhino.

## Methods

TEPEAK (Fig 1) is a modular workflow for detecting, classifying, and analyzing polymorphic TE insertions across population-scale genomic datasets. The pipeline can operate in several input modes, allowing flexibility depending on data

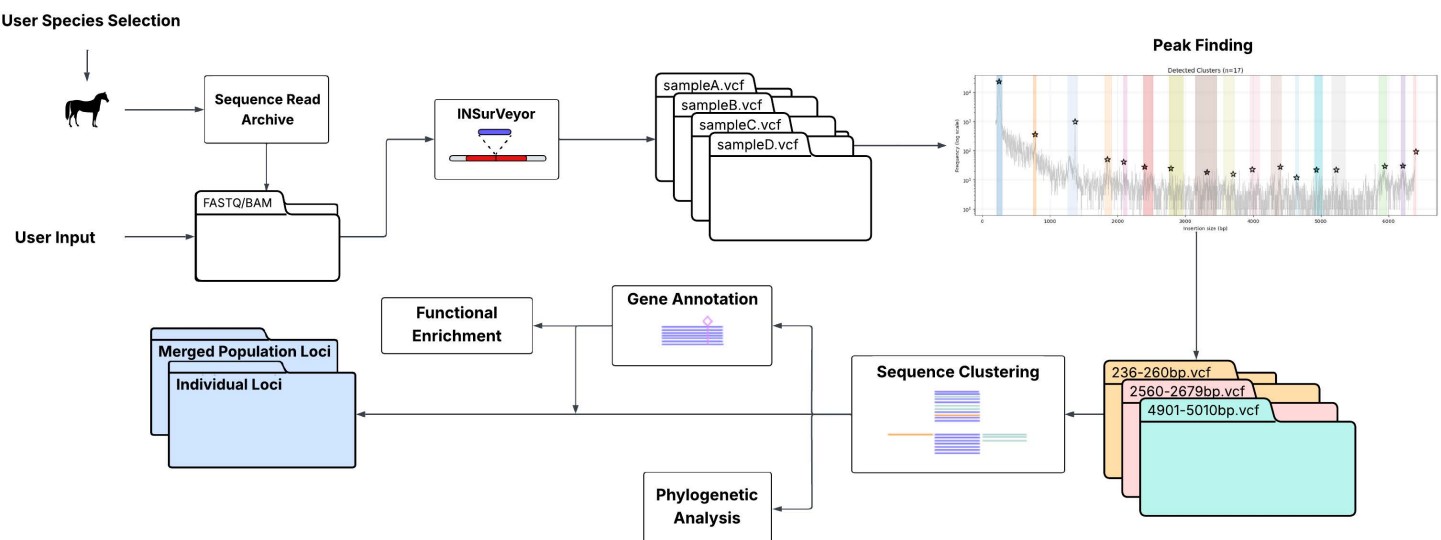

**Fig 1. Overview of the TEPEAK pipeline.** Inputs (SRA accessions, FASTQ, or BAM/CRAM) are processed to call insertions per sample, then pooled to detect size-enriched peaks indicative of TE families. Sequences within peaks are clustered to form family representatives and annotated, after which TE loci are extracted across individuals, merged into a population-level table, and intersected with gene features. Optional modules support functional enrichment, phylogenetic summaries, and population structure visualizations. Outputs include per-sample calls, cohort-level loci and annotations, and summary plots. Images by OpenClipArt.

availability and preprocessing requirements. All stages of execution are managed within Snakemake, ensuring reproducibility and consistent environment control [40].

## Data collection and preprocessing

The workflow begins with user-supplied input in one of three formats: a list of SRA run accession numbers, paired-end FASTQ files, or pre-aligned BAM/CRAM files derived from Illumina short-read sequencing data.

In the simplest configuration, users provide a plain text file containing SRA accession numbers to be analyzed. Optionally, an accompanying SRA RunInfo CSV (exported from the NCBI Sequence Read Archive) may be supplied [41]. While not required, this file includes additional metadata—such as *Bases*, *LibraryLayout*, *Platform*, and *Organism*—that can be retained for downstream comparative or population-level analyses.

Each accession is automatically retrieved using the SRA Toolkit (fastq-dump), converted into paired-end FASTQ format, and aligned to the user-provided reference genome with BWA-MEM v0.7.17 [42]. In addition to SRA-based execution, we validated TEPEAK's local FASTQ input mode using user-supplied paired-end FASTQ files in an end-to-end test case, as documented in S1 Protocol. We recommend the use of a chromosome-level reference assembly, as fragmented or lower-contiguity assemblies can reduce the accuracy of structural variant and TE insertion detection, complicate locus merging, and limit functional annotation precision.

After insertion calling on each sample, TEPEAK's peak-finding module identifies statistically enriched insertion-size intervals across the population. Consensus sequences from these peaks are automatically compared against the Dfam transposable element database [11] to assign family-level classifications. Users may also specify custom insertion-size ranges, enabling targeted extraction, clustering of highly similar loci, and subsequent gene annotation or phylogenetic analysis. The final outputs include both sample-level and population-merged loci tables with associated annotations.

## INSurVeyor (Insertion calling)

INSurVeyor is a state-of-the-art insertion structural variant calling tool for Illumina short-read sequencing data with sensitivity benchmarks on par with long reads callers [43]. The outcome of this stage is a putative set of TE insertion or SV INDEL sites and sequences for each sample in a Variant Call Format (VCF) file and a summary file detailing the number of insertions per sample. This summary file can be used to further filter poor quality samples.

It is worth emphasizing that TEPEAK's sensitivity for identifying polymorphic insertions derives from the state-of-the-art insertion calls provided by INSurVeyor. INSurVeyor has undergone extensive benchmarking, exhibiting recall and precision on par with or exceeding multiple leading short-read insertion-detection algorithms [43]. Consequently, TEPEAK inherits and leverages this robustness when extracting loci for peak-based clustering. While our peak-finding algorithm refines the resolution of TE insertions, the underlying detection of novel and polymorphic insertions is ultimately driven by INSurVeyor's core strengths. Thus, the overall performance and sensitivity of TEPEAK are closely tied to INSurVeyor's validated accuracy across diverse datasets.

## Peak finding algorithm

The resulting VCF files for each sample are subjected to an adaptive density-based clustering algorithm tailored to identifying TE insertion families across diverse size ranges. Unlike traditional peak detection methods that rely on absolute frequency thresholds, TEPEAK's algorithm employs a local context-aware approach that adapts to the heterogeneous distribution of insertion sizes characteristic of transposable element landscapes.

TEPEAK first constructs an empirical frequency distribution of insertion sizes across all samples within a user-defined size range (default: 100–9000 bp). For each position in this distribution, local insertion density is calculated using a sliding window approach (default: 50 bp window), and compared against a local background estimated from a broader genomic context window (default: 500 bp). Rather than using a global threshold, the algorithm identifies elevated density regions

by computing rolling quantiles (default: 75th percentile) within the local background window, allowing the detection of both high-frequency peaks in common size ranges and lower-frequency peaks in regions with sparser insertion activity.

Contiguous regions exceeding the local density threshold are designated as size clusters, with each cluster characterized by its size range (start-end positions), peak position (local maximum within the cluster), total insertion count, and average density. Clusters separated by fewer than a specified merge distance (default: 100 bp) are consolidated to account for insertion-size variability within the same TE family. This approach effectively distinguishes transposable element activity from background structural variation by identifying "mountains" of elevated insertion density relative to their local neighborhood, regardless of absolute frequency. The algorithm leverages the fundamental observation that elements within the same TE family share characteristic lengths, while also accommodating the reality that different TE families may be present at vastly different copy numbers.

For each detected size cluster, TEPEAK performs hierarchical sequence clustering to identify distinct TE subfamilies that may share similar insertion lengths but differ in sequence composition. Sequences within each size cluster are first extracted, with smart sampling employed for clusters containing more than a specified threshold (default: 200 sequences). The sampling strategy overweights sequences near the cluster's peak position, as these are more likely to represent bona fide TE insertions rather than boundary artifacts or unrelated structural variants.

Extracted sequences undergo a two-stage clustering process optimized for computational efficiency. First, sequences are pre-filtered into length-variant groups (tolerance: ± 10% length variation), exploiting the constraint that highly similar sequences likely have similar lengths. Within each length group, sequences are clustered using a greedy algorithm with pairwise sequence identity thresholds (default: 85% identity). Each sequence is compared to existing cluster representatives using global pairwise alignment (Biopython PairwiseAligner with match score = 1, mismatch/gap scores = 0), and assigned to the first compatible cluster. Sequences failing to match any existing cluster seed new clusters.

This greedy approach reduces computational complexity compared to all-versus-all clustering making the method tractable for large datasets. For the remaining clusters, a consensus representative is selected using one of three strategies: most frequent sequence (default), longest sequence, or sequence closest to the median length. These options allow users to tailor the consensus to their analytic goals: the most frequent sequence emphasizes population-level commonality, the longest sequence preserves potential structural completeness for downstream annotation, and the median-length representative provides a balanced profile for comparing families with heterogeneous insertions. This flexibility enables analysts to interrogate both dominant and rare sequence variants, improving interpretability across evolutionary or functional contexts.

When the number of sequences in a size cluster exceeds the sampling threshold, cluster sizes are extrapolated to estimate total insertion counts across the full dataset. This hierarchical clustering approach enables TEPEAK to distinguish multiple TE subfamilies or paralogs that may occupy overlapping size ranges, a common scenario in eukaryotic genomes with complex TE landscapes.

### Representative sequence selection and TE family annotation

For each identified sequence family, the consensus representative sequence is automatically queried against the Dfam transposable element database [11]. The API returns potential matches ranked by e-value, bit score, and alignment quality. TEPEAK records the top hit's Dfam accession, TE superfamily classification, family name, and statistical support metrics (e-value, bit score) in the output annotation table.

The integration of adaptive density-based size clustering with hierarchical sequence clustering and database annotation enables TEPEAK to systematically characterize TE family composition across populations while maintaining computational tractability. The output includes per-family statistics (size range, estimated copy number, representative sequence, Dfam classification) that serve as input for downstream functional annotation, population genetic analyses, and comparative genomic studies.

## Locus extraction, intersection, and downstream annotation

Once peaks and corresponding TE families have been identified, TEPEAK extracts all insertion loci belonging to each peak size range and merges them across individuals to produce a population-level locus table. Each entry represents a shared or unique insertion locus, maintaining sequence-level information for each contributing sample, while sequences exceeding 10 kb are truncated or flagged, as the Dfam API imposes sequence length limits.

These loci are then intersected with the user-provided gene annotation (GTF) to determine overlap with exons, introns, untranslated regions, or intergenic space. The output includes per-locus annotations specifying the genomic feature type and, when applicable, the associated gene ID. This enables downstream functional analyses and the identification of TE insertions potentially affecting coding regions or regulatory domains.

## Additional TEPEAK analyses

Several optional modules extend TEPEAK's functionality beyond core insertion discovery and annotation:

- **Functional Enrichment Analysis** – The sets of genes intersecting with TE insertions can be analyzed for enriched Gene Ontology (GO) categories or KEGG pathways via g:Profiler [44].

- **Phylogenetic Reconstruction** – For the most abundant TE families in the dataset, representative consensus sequences are aligned using MAFFT (with the --auto mode for automatic algorithm selection) and subjected to approximate maximum-likelihood phylogenetic analysis using FastTree [45,46]. The pipeline automatically selects the top annotated TE families for analysis, constructing phylogenetic trees to infer evolutionary relationships among TE subfamilies within the dataset. FastTree is run with the Jukes-Cantor substitution model and CAT approximation for rate heterogeneity (nucleotide mode), providing computationally efficient tree inference suitable for exploratory analysis of TE family relationships.

Each of these modules is configurable through the TEPEAK configuration file and is disabled by default.

## UMAP dimension reduction analysis

For visualization of breed population structure based on ERE-1 polymorphic insertion patterns, we performed Uniform Manifold Approximation and Projection (UMAP) dimension reduction. We first filtered ERE-1 loci to retain only those with a minimum allele frequency of 0.75 in at least one horse breed, resulting in a binary presence/absence matrix across all samples. UMAP was performed using the umap-learn Python package (McInnes 2018) with the following parameters: n_components = 2, random_state = 42.

## Output

TEPEAK produces a structured hierarchy of outputs, including:

- Per-sample VCFs containing INSurVeyor insertion calls.

- A merged population-level insertion table summarizing shared and unique loci.

- Peak-level size-frequency distributions and plots.

- Dfam-based TE family annotations for representative sequences.

- Gene-intersection and feature-annotation tables.

- Cluster, phylogeny, and functional-enrichment outputs.

Collectively, these outputs provide an integrated view of transposable element activity, population polymorphism, and potential genomic impact within and across species.

## Horse case study

As with many species closely tied to human intervention, horses present a compelling case study for TEs. Horses were domesticated in the Eurasian steppe over 4,000 years ago. The evolution of horses, owing to their pivotal role in human civilization, has left a myriad of genetic footprints influenced by various historical and cultural inflection points. As the arrival of industrialization initiated the replacement of traditional horse roles, their selective breeding was intensified, particularly for those horse breeds used for specific purposes like racing in western cultures. This has resulted in a spectrum of modern horse populations, ranging from those that have undergone extreme genetic interventions to those that experience less structured breeding and life histories. While TEs' fundamental role in genome evolution is appreciated, the interplay between TEs and the genetic repercussions of diversity loss, especially in species with a varied breeding history, remains poorly understood.

To demonstrate TEPEAK, we ran it on 257 horse samples across 11 breeds, Thoroughbred, Quarter horse, Friesian, Hanoverian, Freiberger, Arabian, Mongolian, Akhal-Teke, Jeju horse, Tibetan, and Standardbred (Fig 2, Table 1), which represent a basal subset of the over 600 horse breeds. This selection was also made in part on the availability of SRA accessions on NCBI. Accessions were selected based on sequencing depth and metadata completeness, prioritizing Illumina paired-end WGS libraries with approximately 100M read pairs and 40 G bases per run. These breeds offer diverse ancestries and histories, ranging from well-documented breeding lineages to ones whose origins are subject of debate. Geographically, these breeds offer a broad representation of environmental and cultural influence. Most importantly, these breeds represent a wide spectrum of human influence on horse genetics.

We selected 8 varieties of contemporary horses and three Asian landraces (Mongolia, Tibet, and Korea). Of these, several are subjected to high levels of contemporary anthropogenic selective pressure for racing and dressage [47–52,54,56]. Others, such as pastoral horses from Mongolia and Tibet, may have much lower levels of human intervention in their breeding. We expect some geographic structuring in relatedness across these varieties, with horses from Korea, Tibet,

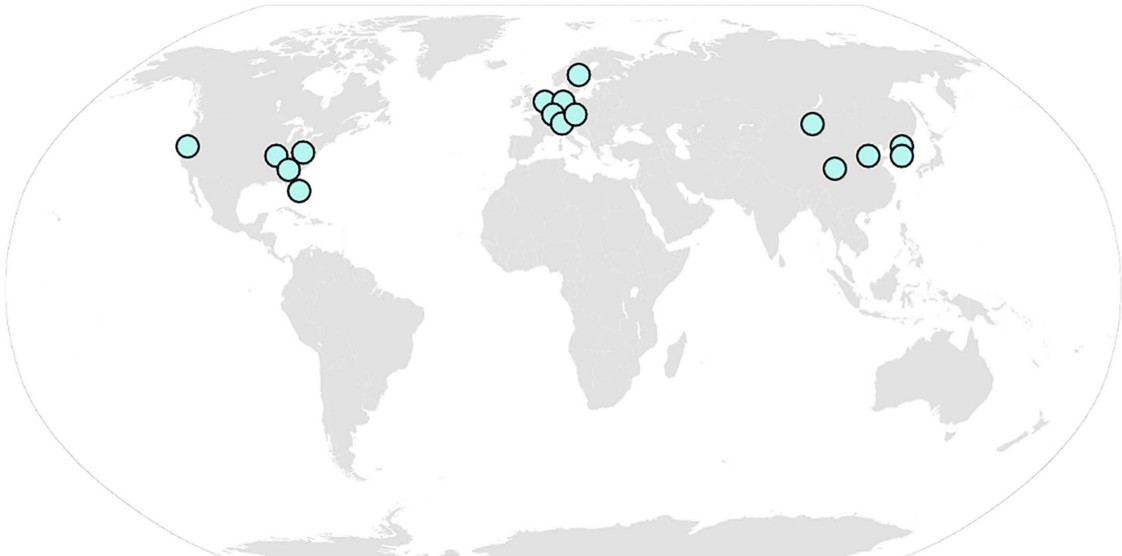

**Fig 2. Geographic locations of institutions that conducted sequencing for the horse samples analyzed in this study.** Samples were obtained from NCBI SRA sequence repositories (accessions listed in S2 Table). Map locations do not reflect sample collection localities. Samples and source research projects were chosen to maximize geographical and experimental design diversity. Map obtained from Wikimedia Commons (https://commons.wikimedia.org/wiki/Maps_of_the_world#/media/File:BlankMap-World.svg), created by Canuckguy and contributors, released into the public domain.

**Table 1. Summary of the 11 horse groups included in this study, chosen to represent a broad range of geographic origins, breeding histories, and selection pressures. Each entry highlights key historical context, characteristic traits, and a potential primary form of human-driven or environmental selection shaping the group's genetic background.**

| Group | History | Characteristics | Genetic Pressure |
|---|---|---|---|
| Thoroughbred | Developed in 17th-18th century England by crossing native mares with imported Arabian and Turkoman stallions [47]. | Known for speed and stamina, most successful racing breed. Experienced a very strong selection in a short period of time. | Racing |
| Quarter Horse | Originated in 17th century America, descended from English horses brought by colonists and influenced by native horses. Currently the largest horse population in the U.S [48]. | Known for speed in short distances, versatility, and strength; widely used in rodeo events. | Racing, rodeo |
| Friesian | Originated in Friesland, Netherlands, dating back to the Middle Ages. Used for war, carriage, and farm work [49]. | Recognized by its black coat, powerful build. | Dressage, show |
| Hanoverian | Originated in Hanover, Germany, and developed as a carriage and military horse. More recently selectively bred for athleticism and dressage. Cross bred with Thoroughbreds for agility in the 19th century [50]. | Versatile and athletic, commonly used in Olympic dressage, jumping and other equestrian sports. | Dressage |
| Freiberger | Originated in Switzerland, primarily developed from imported warmblood breeds as artillery draft horses. Very small population with a closed stud-book existing since the late 20th century. Believed to have been influenced by Friesian, Arab, Thoroughbred, and Hanoverian prior to the stud-book [51]. | Versatile, primarily used for riding. | Agriculture, recreation |
| Arabian | One of the oldest horse breeds, originated in the Arabian Peninsula, highly prized for stamina and beauty [52]. | Distinct head shape, high tail carriage, known for endurance and heat tolerance. | Show, breeding |
| Mongolian | The oldest and most genetically diverse population. Estimated adoption into East Asian nomadic societies ca. 1200 BCE [53]. | Stocky build, short legs well adapted to extreme climates. | Agriculture |
| Akhal-Teke | Originated in Turkmenistan, known for ancient lineage whose origin is of much debate [54]. | Recognized for its metallic sheen, endurance. | Show, conservation |
| Jeju Horse | Native to Jeju Island, South Korea, whose origins are debated. Historical evidence of Mongolian horse introduction in the late 1200's. However, archeological evidence suggests the presence of horses 2500 years ago. [55]. | Small and stocky build, used in agriculture. | Agriculture, conservation |
| Tibetan | Introduced to the Tibetan plateau in the first millennium BCE. to the Tibetan Plateau, adapted to high altitudes [19]. | Stocky build, short legs. | Agriculture |
| Standardbred | Developed in 19th century America, initially from the Darley Arabian line of Thoroughbreds. And later influenced by Norfolk Trotter and Morgans. Historical lineage is better documented than many other popular racing breeds as a whole [56]. | Heavier build than Thoroughbred. Known for its ability to maintain "standard" trot; widely used in harness racing. | Racing, show, recreation |

and Mongolia likely sharing an initial history of dispersal after the initial domestication of the horse in the late 3rd or early second millennium BCE [57–59].

Prior to alignment, no external preprocessing was performed. All standard QC and mapping-based filters are applied within the pipeline. Each sample was aligned to the latest available horse genome assembly, EquCab3.0, which was isolated from a thoroughbred [60]. The resulting aligned files (BAM format) were input into INSurVeyor. Each sample's VCF file was subjected to filtering to exclude insertions with a size below 100 bp. Those samples with less than 1000 insertions were rejected. The resulting mean insertion count per sample was 4312. The peak-finding algorithm identified prominent local maxima at 246 bp, 368 bp, 490 bp, 651 bp, 781 bp, 873 bp, 1,374 bp, 6,400 bp, 6,603 bp, 6,995 bp, and 7,112 bp (Fig 3). The multiple maxima observed in the 6–7.5 kb range reflect closely spaced peaks within a region of elevated insertion density, rather than implying that each maximum represents a fully independent insertion class. To characterize the identified peaks, a representative consensus was established for each peak using the Dfam database. Dfam results annotated five instances as structural components of LINEs (Long Interspersed Nuclear Elements), two instances as

PLOS Computational Biology

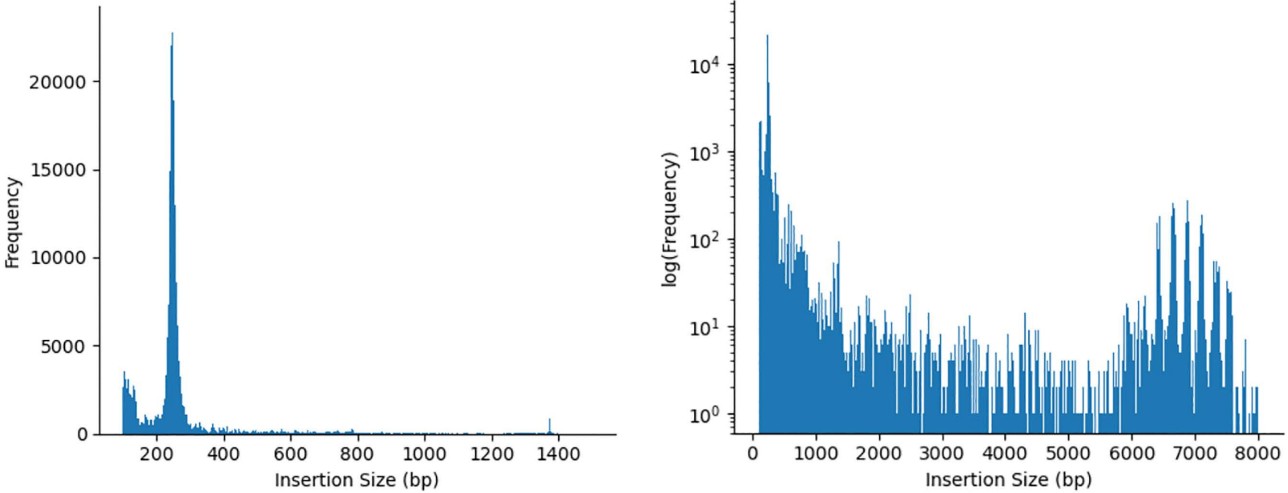

**Fig 3. TEPEAK's size-frequency distribution for 257 horse genomes with a linear- and log-scaled y-axis.** Local maxima were identified at 246 bp, 368 bp, 490 bp, 651 bp, 781 bp, 873 bp, 1,374 bp, 6,400 bp, 6,603 bp, 6,995 bp, and 7,112 bp. Multiple local maxima in the 6–7.5 kb range reflect closely spaced peaks within a region of elevated insertion density. All maxima are reported for completeness; subsequent analyses focus on peaks selected as illustrative examples (ERE-1 and ERV2).

LTRs (Long Terminal Repeats) from ERV1 (Endogenous Retrovirus 1), and four instances without existing annotations. For the downstream case study, we chose to focus on peaks that yielded clear family-level annotations and interpretable population-level patterns (ERE-1 and ERV2).

To further investigate peaks not annotated in Dfam, two separate approaches were undertaken. First, the sequences corresponding to these unannotated peaks were subjected to BLAST against the horse genome. In the case of the 246 bp peak, BLAST identified the peak consensus against the consensus sequence for Equine Repetitive Elements 1 (ERE-1) (94% PID, 92.56% coverage). BLAST did not return any annotations for the peaks at 651 bp, 873 bp, and 1,374 bp.

Second, to address the absence of annotations for the peaks at 651 bp, 873 bp, and 1,374 bp, we used BLAT (BLAST-Like Alignment Tool) from the UCSC (University of California, Santa Cruz) Genome Browser. While BLAT recovered annotations for these peaks, access to many of its full annotations and consensus sequences were locked behind a subscription service, GIRI Repbase, making it challenging to acquire complete information. Access to this information is not required by TEPEAK, but we aimed to utilize a validated ERV consensus sequence to show TEPEAKs ability to reconstruct full length ERVs as well as identify solo LTRs. BLAT extracted ERV1-LTR annotations for the 651 bp and 873 bp peaks. The full annotations and consensus sequence of the peaks were behind the paywall. The 1,374 bp peak consensus sequence resulted in full coverage for the LTR regions of ERV2. ERV2-LTRs and ERV2-INT's (internal) consensus sequence and full annotations were not behind a paywall.

ERE-1 and ERV2 were selected as the focal points for our case study. TEPEAK was run on 257 horses across 11 groups. Over 1,541,863 insertions (>100 bp) passed INSurVeyor's and TEPEAK's filtering and 35 horse samples failed to reach the minimum number of insertion calls. Insertions located on sex chromosomes were excluded.

## Extraction of ERE-1 loci

The peak finding step of TEPEAK identified a significant peak at 246 bp consisting of 23,092 unique insertions. BLAST database query confirmed the insertions at this size to be a subfamily member of the perissodactyl-specific SINE family of Equine Repetitive Elements (ERE), ERE-1. To date ERE consists of four major subfamilies: ERE-1, ERE-2, ERE-3, and ERE-4 with consensus sequences ranging from 228 to 268 bp [61].

Analysis of the size-frequency histogram showed the ERE consensus range proxies roughly 90% of prominent sizes immediately adjacent to the 246 bp peak, ranging from 220-280 bp. Extraction of all insertions (526,675) in the 220–280 bp range and filtering of sequences using each subfamily's consensus sequence confirmed the presence of all four ERE sub-families. Previous work has shown ERE-1 to be the most polymorphic subfamily with strong evidence that ERE-1s were the most recently active element in the horse genome [62–66]. Our results agree with these findings. ERE-1 insertions are not only more polymorphic in terms of interbreed and population frequency variance, but also sequence divergence and locational preference (Fig 4).

### Extraction of ERV2-LTR and full-length ERV2

Queries of the consensus sequence at the 1,374 bp peak against the Dfam and BLAST returned no closely match-ing annotations. It was initially believed that this TE was a novel discovery. However, further research revealed the annotations to be owned by a subscription service, GIRI. Utilizing UCSC [67], which has access to the GIRI RepBase, to query the 1,374 bp consensus sequence against the horse reference confirmed this TE to be the long-terminal repeat region of an endogenous retrovirus, specifically ERV2-1N-EC_LTR (ERV2-LTR) [60,61,68]. ERV's contained open reading frames (three genes) when initially infecting hosts millions of years ago. This internal region is flanked by identical long terminal repeats (LTRs). Over time the internal region becomes inert and often becomes excised through recombination leaving the two LTRs in close proximity to each other [3, 69]. It is estimated that 90% of all ERVs are solo LTRs [3,70].

We then demonstrated TEPEAK's ability to extract full-length and intact ERVs without prior knowledge of the inter-nal region sequence. While a full-length ERV may be observed as a significant peak in the size-frequency histogram, the polymorphic nature and similar size to much more common LINE insertions make it difficult for de-novo extraction as was performed on ERE-1. To demonstrate this we first extracted and filtered all ERV2-LTR loci by pairwise align-ment with a consensus sequence constructed from all insertions in the 1,370–1,380 bp range. Any sequence with less than 85% PID was rejected. This filtering extracted 2,483 insertions over 544 loci. Subsequently, we merged all loci within a distance of 10,000 bp to one another. This threshold was a deliberate overestimation for the interior region length in order to account for instances where paired LTRs might not have been successfully identified as well as mit-igating any other resolution inconsistencies in repetitive regions. This merging filter reduced the number of loci to 200. It is worth noting the average distance of the original loci prior to merging was less than 50 bp. The merged loci were then intersected against the entire population's insertions whose size was greater than 4,000 bp. This threshold is a conservative underestimate for the internal region of the ERV. Finally, the resulting sequences were split in half where each half was scanned against the ERV2-LTR consensus sequence. A successful match in both halves was classified as full length ERV.

### Results

#### Polymorphic insertions of ERE-1 shows distinct breed population structure

As ERE-1 represents the most recently active ERE element we chose to focus on just ERE-1's population structure. We thus isolated insertions with sequences that share at least 85% identity to the ERE-1 consensus. This filtering extracted 22,451 loci and 285,828 insertions. Across all breeds, a majority of ERE-1 loci are rare (less than 0.1 AF), 2,900 loci were in at least 50% of the population in any breed, and 116 loci were in at least 50% of all breeds. UMAP results validate our intuitions for most breeds (Fig 5B). Notably, the Jeju horse, isolated on an island and not subjected to racing or dressage breeding pressure, serves as a pseudo control group.

Thoroughbred and Arabian breeds exerted a discernible influence on the genomic makeup of Quarter Horse, Stan-dardbred and Hanoverian, as evidenced by their grouping patterns. However, a distinct dissimilarity was observed in the Freiberger population despite the evidence of some shared Thoroughbred lineage. Surprisingly, despite the

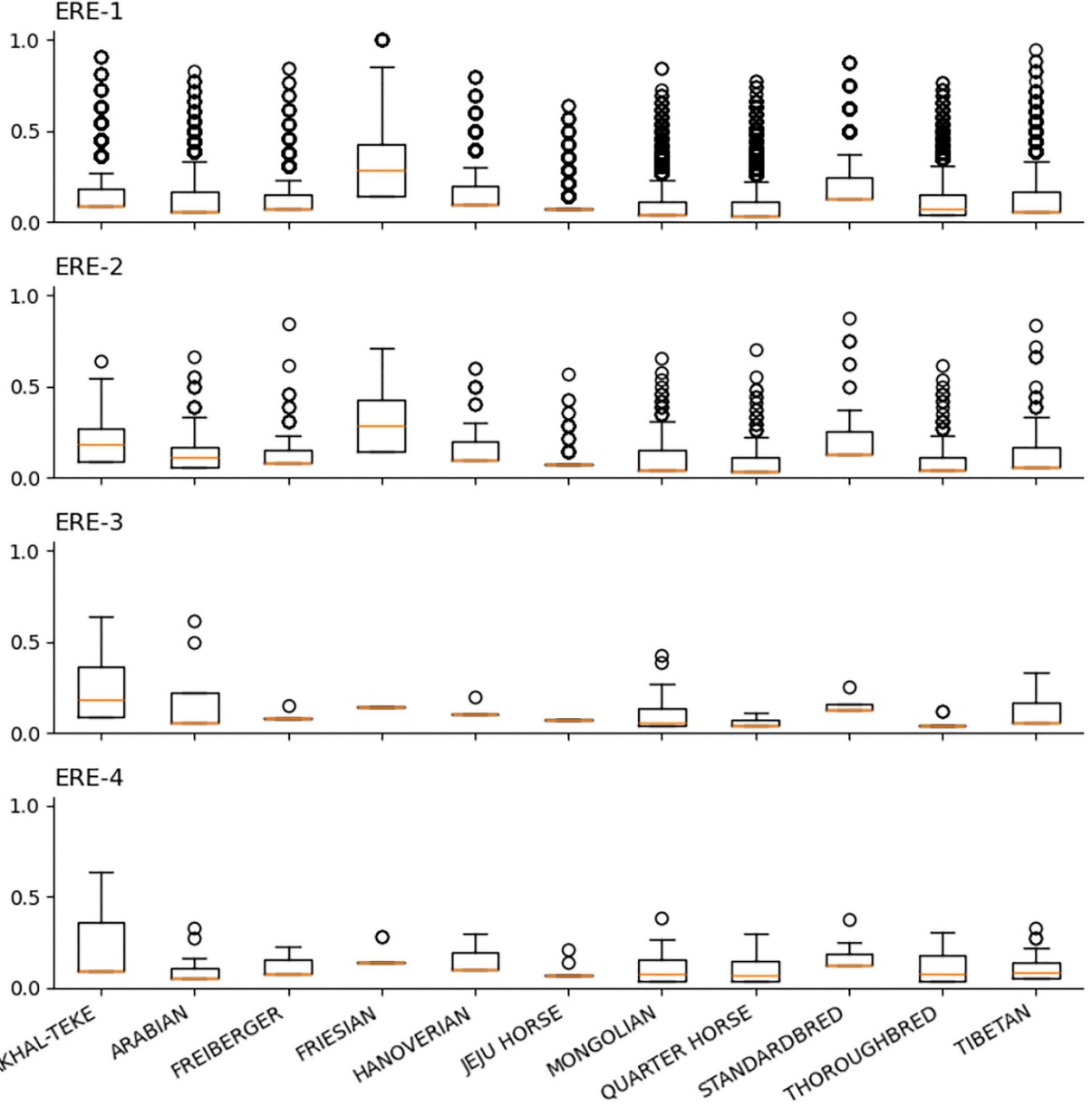

**Fig 4.  Allele frequencies for each of the ERE subfamilies with rare loci removed (less than 0.1 AF).**

influential role of Thoroughbred and Arabian breeds, both exhibited relatively few nearly fixed (> 0.75 AF) ERE-1 loci, appearing to have passed on only a limited subset of nearly fixed loci. To further understand this subset of shared loci, we filtered all loci shared by at least 75% of individuals within each breed (Thoroughbred, Arabian, Standardbred, Quarter Horse) and annotated their sites for genomic features. The initial filtering extracted 88 loci, 27 of which intersected with genomic features (10 intronic regions, 4 exon regions, 12 unknown horse genes, and one enhancer region (Table 2). Curiously, 25 of these loci are also nearly fixed in the Mongolian population. The intersected genes highlight critical developmental, behavior, and physiological processes associated with memory, temperament,

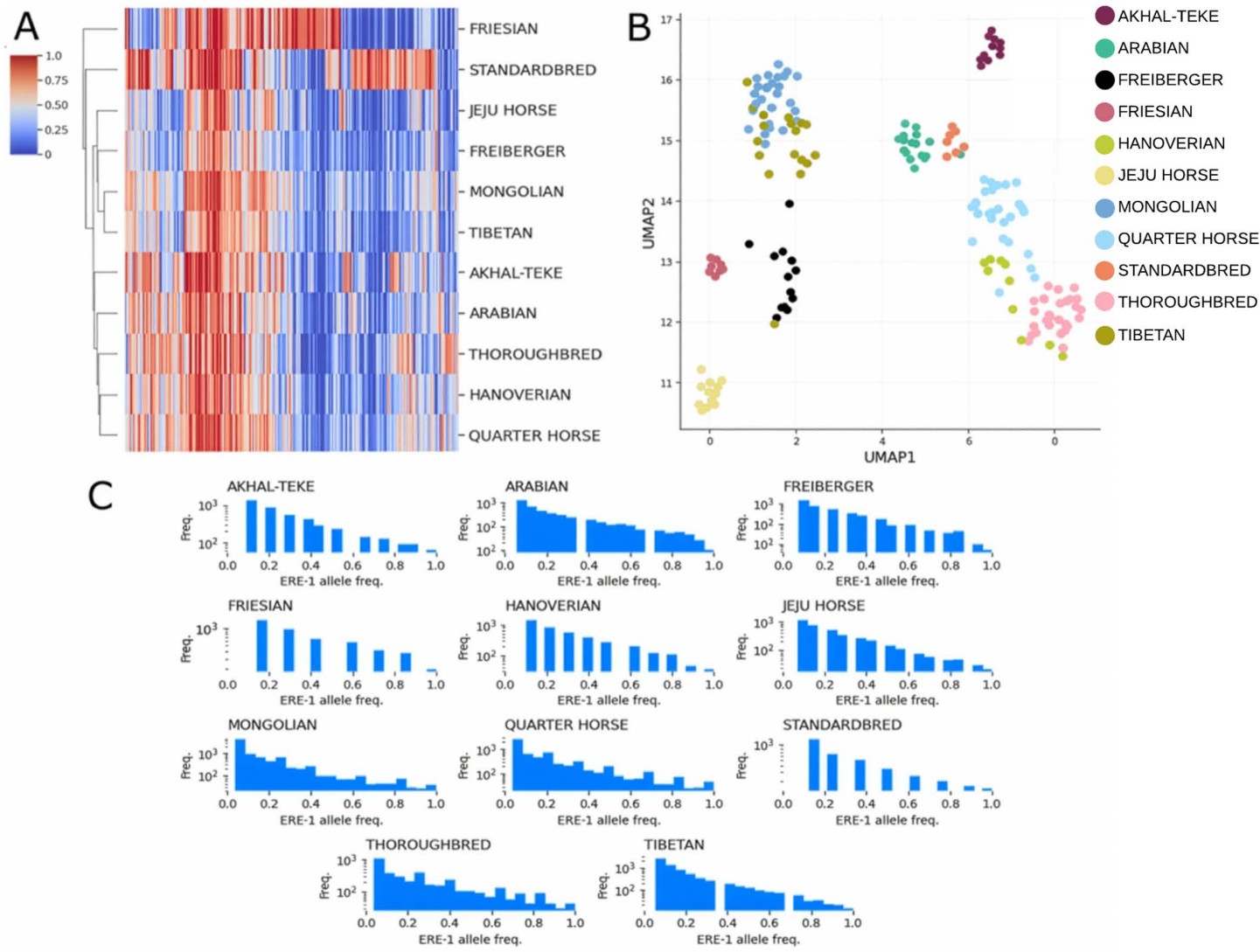

**Fig 5. A. A heatmap of the allele frequencies of individual ERE-1 insertion loci (x-axis) across the horse breeds (y-axis).** Only loci with a minimum allele frequency of 0.75 in at least one of the horse breeds are included. Color scale indicates allele frequency in each breed, the transition from blue to red indicates ERE-1 insertions in the majority of samples for the respective breed. The y-axis represents the resulting dendrogram from hierarchical clustering. B. Visualization of UMAP dimension reduction on allele frequency data based on alleles that are present in at least 0.75 of one horse breed. C. Inter-breed allele frequency distribution for all ERE-1 loci present in at least one sample belonging to the respective breed.

neurological and vascular systems, as well as major biological metabolic and synthetic pathways. We acknowledge insertions into introns may be nonfunctional unless demonstrated otherwise experimentally. However, variation in heritability of these shared loci especially in breeds with drastic selective breeding pressures is notable due to the greater chance of viable and stable insertions [5]. This notion also demonstrates TEPEAK's ability to filter for genomic targets in additional analysis.

There is a direct correlation between breeds subjected to selective breeding pressure and the prevalence of both rare and fixed loci. This relationship highlights the genetic loss that results from such practices. Intriguingly, the Quarter Horse appears to deviate from this trend. This possibly could be attributed to the high introgression of other lineages as well as

**Table 2. Genes associated with the nearly fixed loci shared among Mongolian, Thoroughbred, Standardbred, Quarter Horse, and Arabian breeds. All related traits sourced from GeneCards unless specified otherwise.**

| Gene name | Related traits | Insertion type |
|---|---|---|
| ALDH18A1 | Temperament [71] | Intron |
| KIF11 | Spindle Dynamics/Dendrites<br>Overexpression believed to offset Alzheimers in humans [72] | Exon |
| ALK | Insulin receptor/brain development | Exon |
| SYN3 | Synaptogenesis/modulation of neurotransmitters | Intron |
| Enhancer for MEIS1 | Hox Gene/head-tail axis development | N/A |
| ZNF121 | Predicted DNA binding TF activity | Intron |
| Unknown Horse Gene ENSECAG00000057187 | Unknown | Unknown |
| MyH10 | Cell maintenance with roles in coronary, vasculature function [73] | Intron |
| GUCY1B2 | GTP and heme binding | Exon |
| Rag1 | Immunoglobulin V-D-J recombination | Intron |
| AUTS2 | Neurodevelopment, candidate gene for numerous neurological disorders | Intron |
| DCLK1 | Memory and general cognitive abilities | Intron |
| LRP1B | Clearance of extracellular low density lipids | Intron |
| PALLD | Actin cytoskeleton | Intron |
| IGF2 BP3 | RNA synthesis - late development | Intron |

its extensive population in the United States. The Quarter horse's connection to Spanish, Native American, and Anglo breeds, many of which are not explored in this study, suggests unique genetic dynamics from the different populations and encourages a future exploration.

## Mongolian horses as a source of ERE-1 alleles

The Mongolian breed is the oldest and most genetically diverse contemporary horse population [74]. It is thought that a majority of SINEs amplified at least 30 million years ago in proto-ancestors of mammals [75,76]. Although ancient DNA suggests that horses were first domesticated in western Eurasia, early horses dispersed to the region at least as early as ca. 1200 BCE [53] with Mongolia likely playing a key role in the initial dissemination of horses into nearby regions of East Asia [77]. Beginning in the first millennium BCE and continuing through the Middle Ages, expansive pastoral empires like those of the Xiongnu, Turkic Khaganate, and the Mongol Empire expanded the role of Mongolian horses across much of Eurasia [78].

The presence of most of the variation in the Mongolian population implies that the ERE-1 loci were extant at the time of breed diversification. It is also apparent that this variation continues to exert an influence on modern-day horse genetics despite strong selective breeding in many populations.

The Mongolian breed exhibits the greatest amount of fixed and rare ERE-1 loci, highlighting the intuition that the Mongolian breed is the best representation of ERE-1's impact on an intact population. To further investigate the shared relationship between Mongolian and other modern-day breeds, we assessed the number of shared loci in pairs of horse breeds, considering instances where at least 75% of samples in each breed exhibited insertions at each loci (Fig 6). These results affirm the Mongolian breed as harboring the most diverse set of loci. Furthermore, we observed a notable pattern in the distribution of shared loci, particularly among breeds in proximity to the Eurasian steppes.

A considerable amount of shared loci between Mongolian and Akhal-Teke was also observed. These results point to shared history between the two varieties, and encourage further exploration into ancient connectivity across Inner Asia incorporating ancient genomes.

PLOS Computational Biology

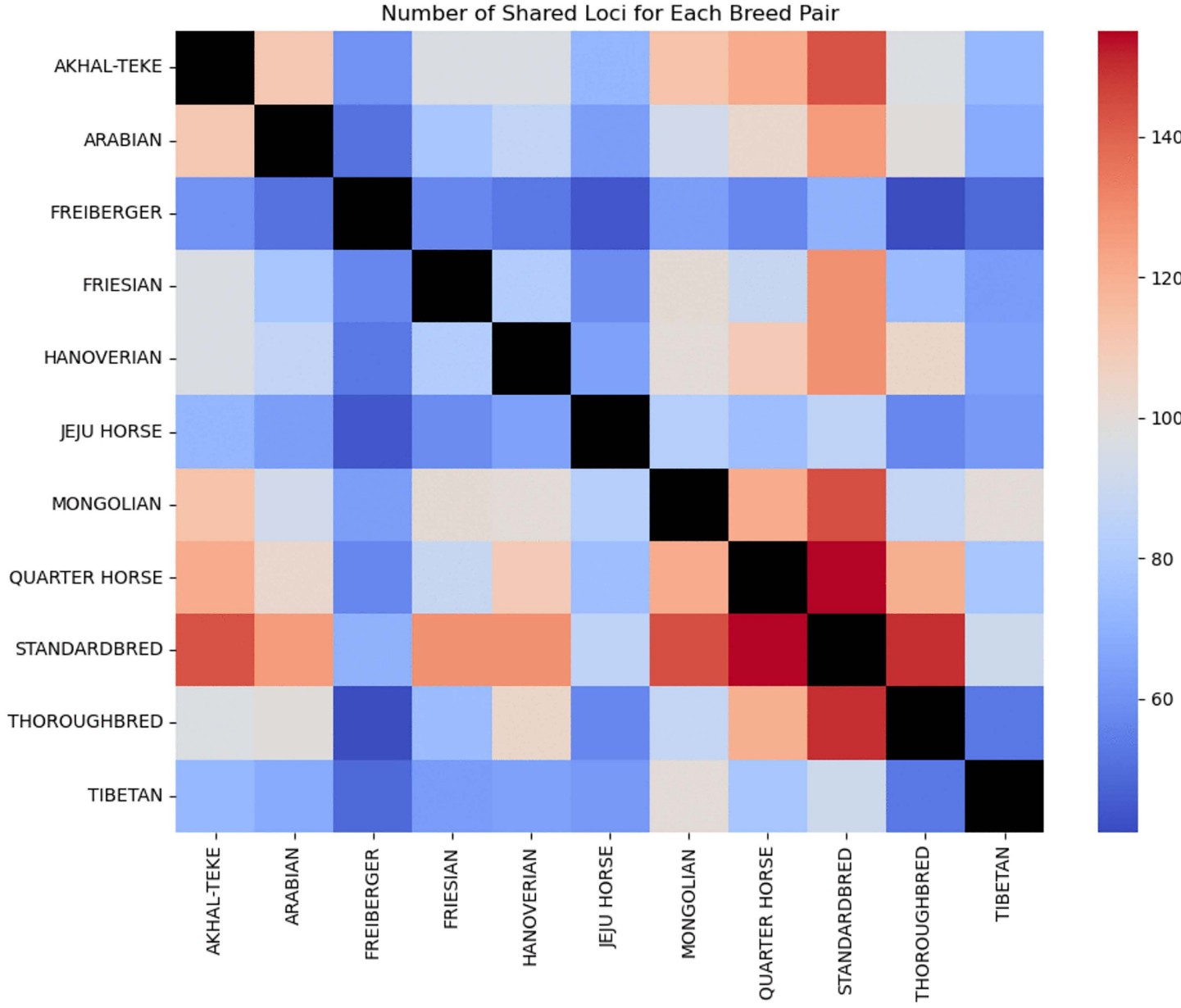

**Fig 6. The number of pair-wise shared high-frequency (AF>=0.75 in both groups) loci.**

As discussed in the preceding section, we emphasized the limited number of nearly fixed loci shared among racing and Mongolian horses, which are closely tied to crucial genes. Antithetical to this we observed a number of loci that are variable in Mongolian populations, but have become fixed in selective breeds. Here we explore the number of nearly fixed loci shared among racing breeds, but not Mongolian. This filtering extracted 2 intron and 1 exon insertions into different genes (Table 3). The intron loci relate to bone and growth development regulation and the exon loci to antigen production. The loci that are polymorphic in Mongolian and Tibetan breeds and fixed loci in nearly all race breeds could be explained by a founder effect and selective breeding.

**Table 3. Genes associated with the polymorphic loci in Mongolian and Tibetan, but fixed in Thoroughbred, Standardbred, Quarter Horse, and Arabian.**

| Gene | Mongolian | Tibetan | Akhal-Teke | Thoroughbred | Standardbred | Quarter Horse | Arabian |
|------|-----------|---------|------------|--------------|--------------|---------------|---------|
| BBX | 0.38 | 0.28 | 1.0 | 1.0 | 0.8 | 0.85 | 0.84 |
| BTN1A1 | 0.42 | 0.43 | 0.36 | 1.0 | 0.86 | 0.85 | 0.76 |
| PTPN2 | 0.42 | 0.28 | 0.82 | 1.0 | 0.8 | 0.81 | 0.76 |

### Genomic impact/Functional enrichment of ERE-1

Next, in order to highlight TEPEAK's ability to gauge the overall genomic influence of a TE, we performed an annotation and functional enrichment analysis of all ERE-1 loci. The results of this intersection pinpointed the presence of ERE-1 in 56 coding DNA sequences (CDS), 5,785 intronic regions, and 539 5' or 3' UTRs. 55.3% of these genes have been subject to two different ERE-1 loci. Of the CDS loci several significant genes were noted including collagen production and joint health (COL1A2), skeletal muscle development (WASHC5), significant regulator of Hox expression-vertebrate trunk development (Nr6a1), anti-bacterial defense (C8A), smooth muscle vasoconstriction (EDNRA), neuroprotein degradation (UBQLN1), epithelial sodium channel regulation (CNKSR3), and various DNA and RNA processing and regulation (BTG4, ATG14, WDR75, TBCD, DTWD2, DDX31) [79].

We subsequently performed functional enrichment analysis on all the CDS and exon intersected genes. ERE-1's presence is notably pronounced in several cellular and biological pathways [80] (Fig 7).

We also aimed to explore the potential for evolutionarily conserved relationships of these functional loci across three breeds: Mongolian, Quarter Horse, and Thoroughbred (Fig 8), chosen because of their large sample size, distinct UMAP clusters, and varying degrees of perceived selective breeding influence. Preliminary observations suggest that, despite intronic regions being conserved, the Mongolian loci experienced much higher rates of fixed intronic loci (>200) compared

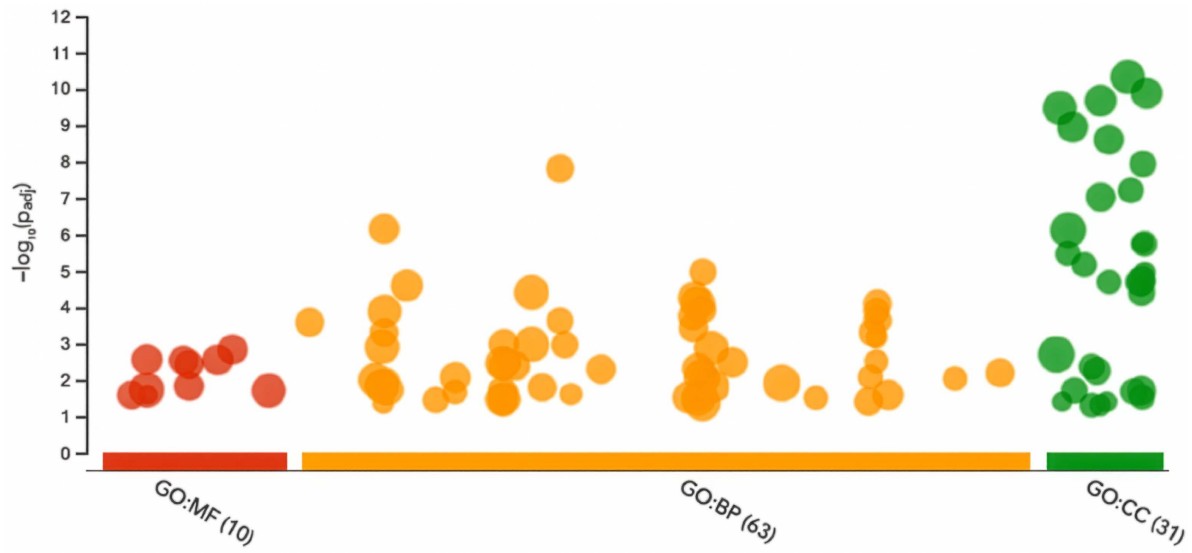

**Fig 7. Functional enrichment analysis of genes intersected by ERE-1 insertions.** Each point represents an enriched Gene Ontology (GO) term, grouped by ontology category (Biological Process (BP), Molecular Function (MF), Cellular Component (CC)). Within each category, neighboring points can correspond to related sub-ontologies, though the x-axis spacing itself is not quantitative. The y-axis denotes adjusted enrichment significance ($-\log_{10}$ p-value).

to the other breeds which maintained more comparable rates of insertion. Quarter horses had the highest amount of rare and common (0.1-0.5 AF) intronic loci, exceeding Thoroughbred by over 250 loci. These results validate our understanding that intronic insertions are most often highly conserved.

In exonic loci both the Thoroughbred and Quarter Horse displayed matched patterns of insertions, with the Thoroughbred showing higher rates across most genes. The Mongolian horse was observed to have smaller rates of exonic insertions compared to the intronic loci, however displayed a more unique set of loci that appear to be rare or polymorphic in the other breeds. This is undoubtedly an artifact of the Mongolian horse's maintained heterozygosity. In the CDS loci the Thoroughbred was observed to have higher rates of fixed and nearly fixed loci deviating from both Mongolian and Quarter horse in many genes, suggesting another consequence of reduced genetic variation from selective breeding practices.

### Extraction of full-length endogenous retroviruses in horse

TEPEAK was able to extract 31 different full-length ERVs across 10 loci. Multiple sequence alignment of each ERV against RepBase's consensus sequence for ERV2_INT (4,450 bp) validated the presence of the intact internal region in all 31 insertions with no false positives. In order to validate the sensitivity of this method we performed pairwise alignment of ERV2_INT's consensus sequence against all insertions over 4,000 bp (75,623 sequences) [61,81]. This brute-force method is computationally expensive. This method yielded 5 additional full length ERV sequences that matched our previously discovered 10 loci. It also yielded three new loci with three insertions total. Further analysis showed two of the new loci to have valid full-length ERVs. However the third loci, which only contained one full length insertion, contained one LTR region whose sequence shared poor identity to the ERV2-LTR. Furthermore, this loci matched none of the filtered loci extracted from the 1,370 bp-1,380 bp range. Therefore this loci was rejected as not a full length ERV2. Furthermore it was observed that the two new validated loci contained shorter LTR regions (1,290 bp-1,340 bp), which consequently resulted in rejection from the previous method.

We also annotated each of the ERV2 (both LTR and full length) loci for genomic features, resulting in intersections with 112 different genes, 1 CDS, 14 UTR, and 97 intron. It was observed that despite intersecting with critical genes (RPH3A

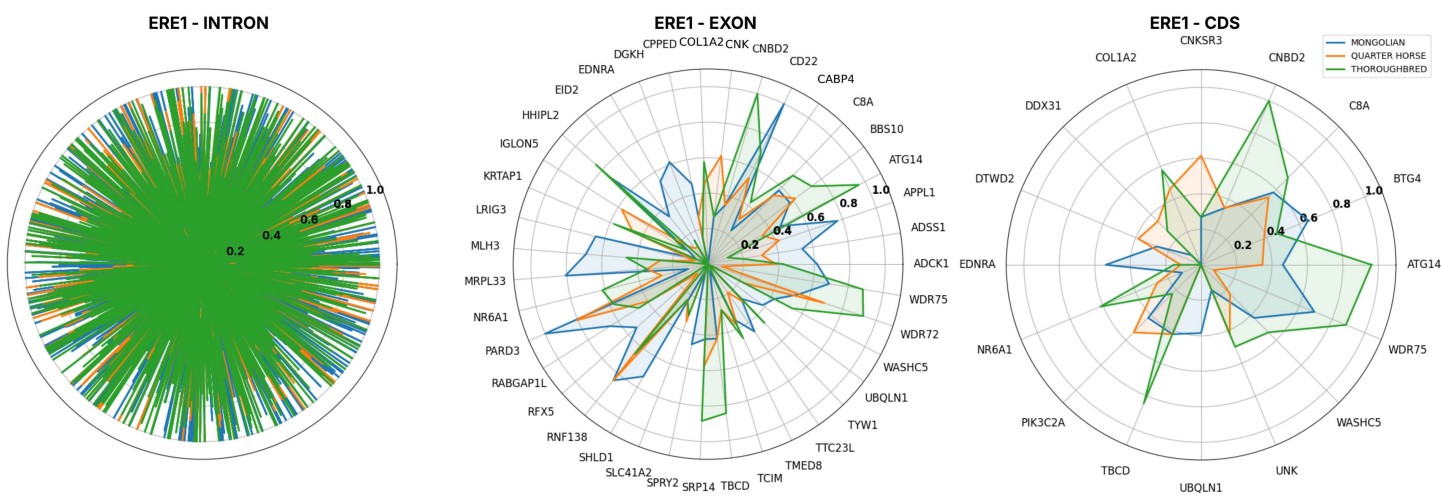

**Fig 8. ERE-1 Allele frequency dynamics across different genomic regions, intron (left), exon (middle), and CDS (right), of three horse breeds.**
X-axis displays human gene orthologs and y-axis breed-specific allele frequency. Intron gene orthologs are omitted for clarity.

(intron), GABRR1 (5'UTR), SVC2 (intron), STXBP5L (intron), and CDK20 (5'UTR)), the loci where full length ERV2 was identified made up a set of the least conserved loci across the species (Fig 9). The loci that make up the most conserved loci across the species intersect with (AVEN (intron), Igλ (intron), HLA (5'UTR), MHC/EQMCE1 (intron), and ZNF77 (5' UTR)). We expect to observe immune system related genes to be enriched with TE activity, however it is notable that the nearly fixed loci for these genes are most present in Eurasian horse breeds (Jeju horse, Akhal-Teke, Mongolian, Tibetan, Arabian).

**Cross-species analysis of equid-specific TEs**

To expand upon our findings from horse genomes and illustrate TEPEAK's flexibility with closely related species, we applied the pipeline to four additional Perissodactyl genomes: black rhino, Brazilian tapir, wild ass, and zebra. These data allow us to assess how frequently equid-specific TEs, identified in the horse cohort, appear in other members of the family Equidae or across divergent lineages. Specifically, we compared insertion sizes around ~200–280 bp (which we initially annotated as ERE-1 subfamilies in horses) to determine whether these sequences could also be recovered in non-equine samples. This cross-species approach highlights TEPEAK's capacity to detect novel TEs in lesser-studied genomes and provides a comparative perspective on the evolutionary distribution of polymorphic elements across different taxonomic groups.

We then filtered the sequences from each of these four species within the 220 bp-280 bp range and constructed consensus sequences. Pairwise alignment of each species' consensus with the horse ERE-1 consensus resulted in >99% for zebra and wild ass. However black rhino and Brazilian tapir were observed to have 76% and 74% PID. We then compared the horse ERE-2, ERE-3, and ERE-4 consensus sequences to the rhino and tapir. In both species the ERE-3/4 resulted in the highest identity, >94%. This further validates the evidence that ERE-1 is the oldest member of the ERE family. We were unable to identify any insertions with high identity (>75% PID) to ERE-1 in rhino or tapir. To our knowledge, wild ass is the only species tested with annotations for this particular sequence, labeled as ERE-1 (Table 4).

Despite the limited insertions in the range of ERV2-LTR, we were able to identify sequences with high PID (>98%) in both wild ass and zebra, but not rhino or tapir. Additionally, due to the low quality reads in these samples no fully-intact ERV2 were able to be extracted. However, BLAT of the horse full ERV2 sequence resulted in sequences with high PID and query coverage, 95% and 92%, in both wild ass and zebra. Neither the LTR nor the full ERV regions are annotated in any database to our knowledge (Table 5).

**Novel TE in black rhino**

Based on the analysis above, we believe that we have identified a novel TE family in black rhino of size 535/536 bp. Due to the proximal nature of insertions in this range, it was hypothesized that these insertions to be ERV-LTRs. With this intuition TEPEAK was able to extract two loci with possible full length ERVs (6705 bp). BLAST results of this sequence showed it to be a Gammaretrovirus (98% coverage, 92.8% PID), validating this TE family to be an ERV.

## Discussion

TEPEAK is a framework designed to identify and characterize TEs in populations using concentrations of similarly sized insertions (Fig 3). TEPEAK's strengths lie in its ability to broaden the analysis scope of a project by integrating previously sequenced SRA samples, discovering TEs without prior annotation or sequence knowledge, and aiding users in discerning the population structure and functional impact of the TEs they find. With these capabilities, we expect TEPEAK to be particularly useful for wild populations where much less is known about TE activity and function.

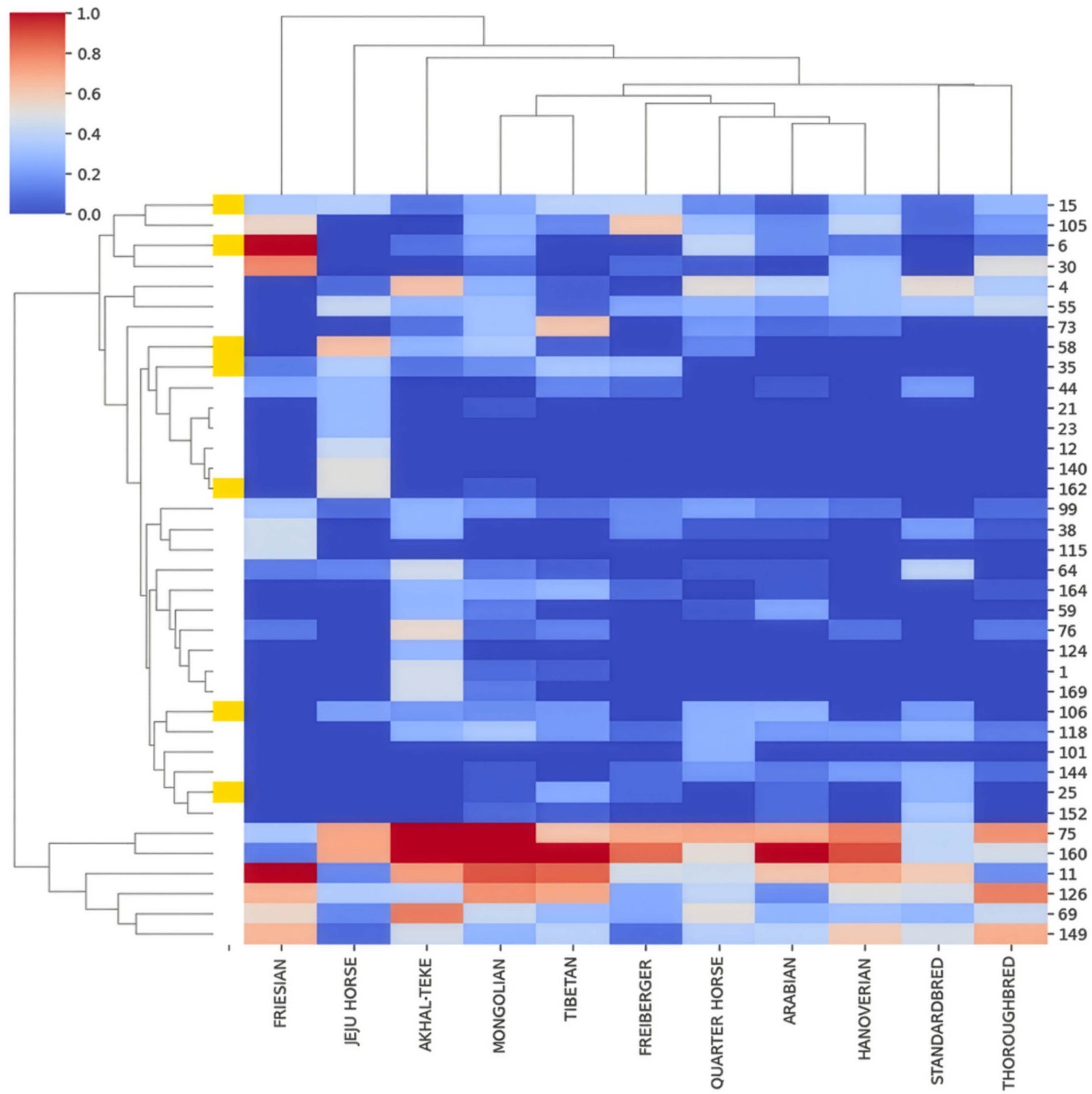

**Fig 9. ERV2-LTR loci's allele frequency in each breed.** Loci that are not present in at least 40% of one breed are excluded. The orange bars on the left mark loci where full length ERV2s were found. Notable gene intersections with loci enriched in the population: 75: EQMCE1/MHC (intron), 160: Igλ (intron), 11: AVEN (intron), 126: unknown horse gene, 69: HLA (exon), 149: ZNF77 (exon).

**Table 4. Pairwise sequence alignment results of each of the horse ERE consensus sequences against the Rhino 220 bp-280 bp consensus sequence. Capital letters indicate matches and lowercase letters indicate mismatches.**

| Horse ERE-1 against Rhino consensus sequence | GGGGCTGGCC | CcGTGGCcgA | GTGGTTAAGT | TCGCgcgctc | cgctgcaggc | 50 |
|---|---|---|---|---|---|---|
| | ggcccagtgt | ttcgttGGTT | CGaATCCTGG | gcgcGGACat | gGCACTGCTC | 100 |
| | ATCAgaCCAc | GCTGAGGCAG | CGTCCCACAT | gCcaCAACTA | GAAGaaccca | 150 |
| | caacgaagaa | tacACAACTA | TGtaccggGG | GGCTTTgGGG | AGAAAAAGGA | 200 |
| | AAAAataaaa | tctttAAAAA | AAaaa | | | |
| Horse ERE-2 against Rhino consensus sequence | GGGGGCCGGC | CcGGTGGCan | AGCGGTTAAG | TtCGCGCGtT | CcGCTtCGGC | 50 |
| | GGCCCggGGG | TTCaCCGGTT | CGGATCCCGG | Gtgcggacat | ggCACCGCTT | 100 |
| | GGCAAaaGCC | ATGCTGTGGt | agGCGTCCca | CATATAAAGT | aGAGGAAGAT | 150 |
| | GGGCAcGGAT | GTTAGCtCAG | GGCCAGTCTT | CCTCAGCAAA | AAGAGGAGGA | 200 |
| | TTGGCAGcaG | TTAGCTCAGG | GcTaATCTTC | CTCAAAAAAA | AAAAAAAAA | |
| Horse ERE-3 against Rhino consensus sequence | GGgGCCAgCC | CAGTGGCaTA | GTGGTTAAGT | TCATGCgCTC | CACTTCAGCG | 50 |
| | GCCTGgGGTT | CGCcGGTTtg | GATCCCgGGC | ACaGACCTAg | GCACCGCTTA | 100 |
| | TCAAGCcATG | CTGTGGCAGG | cGTCCCACAT | ATAAAATAGA | GGAAGATGgG | 150 |
| | CACAgATGTT | AGCTCAGGGC | CAATCTTCCT | CAGCAAAAG | AGGAGGATTG | 200 |
| | GCAgCAGATG | TTAGCTCAGG | GCTAATCTTC | CTCAAAAAAA | AAAAAAAAA | |
| Horse ERE-4 against Rhino consensus sequence | GAGCCAGCCC | TGATGGCCTA | GTGGTTAAAG | TTCGGcGCgC | TCgGCTTcGG | 50 |
| | cgGCCcGGGT | TcGGTTCCcG | GGCgcGGAAC | CACACCACTC | GTCTGTCAGT | 100 |
| | AGCCATGCTG | TGGCGGCgGC | TCACATAGAA | GAACTaGAAG | GACTTACAAC | 150 |
| | TAGAATATAC | AaCTATGTAC | TGGGGCTTTG | GGGAGGaAAA | AAAAAAGAGA | 200 |
| | GAGAGAGGAA | GATTGGCAAC | AGATGTTAGC | TCAGGGcGAA | TCTTTCCCAG | 250 |
| | CAAAAAAAAA | AAAAA | | | | |

**Table 5. Summary of shared equid TE families found in the odd-toed ungulate species and the status of their annotations.**

| | ERV2-LTR | | ERE-1–4 | |
|---|---|---|---|---|
| Species | Present? | Annotated? | Present? | Annotated? |
| Horse | Yes | Yes | Yes, ERE-1–4 | Yes |
| Zebra | Yes | No | Yes, ERE-1–4 | No |
| Wild Ass | Yes | No | Yes, ERE-1–4 | Yes |
| Black Rhino | NO | N/A | Yes, ERE-2–4 | No |
| Tapir | NO | N/A | Yes, ERE-2–4 | No |

To demonstrate TEPEAK and to serve as a guide for other TE explorations, we presented an analysis of TEs in 257 horses across 11 different groups and breeds, which included the related species zebra, wild ass, black rhino, and tapir. Separately, we also ran TEPEAK on giraffes, hippos, panthers, camels, cows, sperm whales, and gray whales (S1 Fig). TEPEAK's SRA interface makes this exploratory, population-scale analysis feasible and efficient, which is critical for fully understanding novel TEs in unstudied populations.

Among the TEs we found in horses, ERE-1, was observed to have a set of fixed loci shared by all groups despite complex history. Despite this shared set, we also identified a number of insertion sites into the coding regions of genes coding for traits such as joint health, skeletal muscle development, and heart function. These results suggest that ERE-1 plays an active role in modern day horse genetics and breeding. We also found ERV2 insertions in both LTR and full-length form that affected immune genes and were nearly fixed in Eurasian horse breeds (Jeju horse, Akhal-Teke, Mongolian, Tibetan, and Arabian).

While TEPEAK is easy to use and powerful, it does require high-quality samples to run effectively, which can be challenging when considering ancient genomes. Currently, TEPEAK only considers Illumina short-read data sets. As long-read sequencing continues to become more popular, this could limit TEPEAK's utility. Our automated identification and

characterization methods can also struggle to correctly separate overlapping families in the same size region and may require some manual curation (e.g., removing LINEs from LTRs).

While the assumption that TE families share characteristic insertion sizes holds true for most element classes, non-LTR retrotransposons such as LINE-1 represent a notable exception. These elements frequently undergo 5′ truncation during reverse transcription, producing insertions with variable lengths that can span several kilobases. TEPEAK mitigates this by identifying statistically enriched size intervals rather than fixed-length peaks, enabling partial recovery of LINE-1 insertions within broader distributions. However, this inherent size variability can reduce peak contrast and complicate precise delineation of individual LINE-1 subfamilies, which we acknowledge as a limitation of the current approach.

## Supporting information

**S1 Table. Consensus sequences for ERE-1, ERE-2, ERE-3, and ERE-4 [61].**
(DOCX)

**S2 Table. All Horse SRA Accession Numbers and their respective breeds used in this study.**
(DOCX)

**S1 Fig. Other species' size-frequency distributions.**
(TIF)

**S1 Protocol. Validation of TEPEAK local FASTQ input mode.**
(DOCX)

**S3 Table. Summarization of outputs generated by TEPEAK during validation of the local FASTQ input mode.**
(DOCX)

**S4 Table. Population-level summarization of outputs generated by TEPEAK during validation of the local FASTQ input mode.**
(DOCX)

## Acknowledgments

We thank members of the Chuong, Taylor, and Layer laboratories for helpful discussions and feedback throughout this project. We are grateful to the researchers who generated and made publicly available the genomic datasets used in this study.

## Author contributions

**Conceptualization:** Devin Burke, Ryan Layer.

**Data curation:** Devin Burke.

**Formal analysis:** Devin Burke.

**Funding acquisition:** Ryan Layer.

**Investigation:** Devin Burke, Ryan Layer.

**Methodology:** Devin Burke.

**Resources:** Devin Burke, Edward Chuong, William Taylor.

**Software:** Devin Burke, Jishnu Raychaudhuri.

**Supervision:** Edward Chuong, William Taylor, Ryan Layer.

**Validation:** Devin Burke, Edward Chuong, William Taylor, Ryan Layer.

**Visualization:** Devin Burke.

**Writing – original draft:** Devin Burke.

**Writing – review & editing:** Devin Burke, Edward Chuong, William Taylor.

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
