## [Decision Letter · Decision Letter 0]

22 Oct 2025

TEPEAK: A novel method for identifying and characterizing polymorphic transposable elements in non-model species populations

PLOS Computational Biology

Dear Dr. Burke,

Thank you for submitting your manuscript to PLOS Computational Biology. After careful consideration, we feel that it has merit but does not fully meet PLOS Computational Biology's publication criteria as it currently stands. Therefore, we invite you to submit a revised version of the manuscript that addresses the points raised during the review process.

Please submit your revised manuscript within 60 days Dec 22 2025 11:59PM. If you will need more time than this to complete your revisions, please reply to this message or contact the journal office at ploscompbiol@plos.org. Please include the following items when submitting your revised manuscript:

We look forward to receiving your revised manuscript.

Kind regards,

Ferhat Ay, Ph.D

Section Editor

PLOS Computational Biology

Ferhat Ay

Section Editor

PLOS Computational Biology

**Journal Requirements:**

At this stage, the following Authors/Authors require contributions: Devin Burke, Jishnu Raychaudhuri, Ryan Layer, Edward Chuong, and William Taylor. Please ensure that the full contributions of each author are acknowledged in the "Add/Edit/Remove Authors" section of our submission form.

5) We have noticed that you have uploaded Supporting Information files, but you have not included a list of legends. Please add a full list of legends for your Supporting Information files after the references list.

Potential Copyright Issues:

i) Figure 1. Please confirm whether you drew the images / clip-art within the figure panels by hand. If you did not draw the images, please provide (a) a link to the source of the images or icons and their license / terms of use; or (b) written permission from the copyright holder to publish the images or icons under our CC BY 4.0 license. Alternatively, you may replace the images with open source alternatives. See these open source resources you may use to replace images / clip-art:

ii) Figure 2. Please (a) provide a direct link to the base layer of the map (i.e., the country or region border shape) and ensure this is also included in the figure legend; and (b) provide a link to the terms of use / license information for the base layer image or shapefile. We cannot publish proprietary or copyrighted maps (e.g. Google Maps, Mapquest) and the terms of use for your map base layer must be compatible with our CC BY 4.0 license.

iii) The following Figure contains a logo or branding: 1. We are not permitted to publish this under our CC-BY 4.0 license, even with permission. We ask that you please remove or replace it.

7) Thank you for stating "Data Used: https://drive.google.com/file/d/1YY-V6zhLOTQsOdQSn2Axqnv-SRHu6i5E/view?pli=1." Unfortunately, google drive does not qualify as an acceptable data repository according to PLOS's standards. At this time, please upload the minimal data set necessary to replicate your study's findings to a stable, public repository (such as figshare or Dryad) and provide us with the relevant URLs, DOIs, or accession numbers that may be used to access these data. For a list of recommended repositories and additional information on PLOS standards for data deposition, please see https://journals.plos.org/ploscompbiol/s/recommended-repositories

8) Please provide a completed 'Competing Interests' statement, including any COIs declared by your co-authors. If you have no competing interests to declare, please state "The authors have declared that no competing interests exist". Otherwise please declare all competing interests beginning with the statement "I have read the journal's policy and the authors of this manuscript have the following competing interests:"

**Reviewers' comments:**

Reviewer's Responses to Questions

Reviewer #1: This looks like a nice package for people to use in curating transposable elements in genome sets and it's generally well-written but there are too many questions about the way the package works to recommend it for publication at this time. It needs a thorough revision, particularly for people who actually want to use TEPEAK. Comments are below.

Minor comments:

Abstract: The github link provided in the abstract redirects to a different page.

Page 5 - "Class II elements move via a “cut-and-paste” mechanism wherein a transposase enzyme directly mediates the joining of the DNA intermediate to its new site" This isn't entirely accurate. Rolling circle transposons are considered Class II but do not use a transposase.

Page 6 - "nearby". This wording assumes the reader is thinking of phylogenetic trees. Probably better to rephrase to "closely related"

Page 18 - Figure 7. I don't know what this figure is attempting to communicate. Why are the dots spaced along the X axis? What does that spacing mean? Why is half of the red bar not occupied by dots? The explanation in the text doesn't tell us much.

Page 19 - Change to "Additionally, this locus..."

More substantial comments, primarily about the methods:

I installed the package and attempted to try it out on some of my own data. This is where several of the comments below arose. The github page provides useful information on how to install the package but that's where the useful information ends. Sooo many questions.

What happens if I want to use the software for a species that does not exist in NCBI or for which there are no SRA entries. Could I use a closely related species? How accurate would it be in that case? There are no explicit instructions on the github page for running my own data. If I have data I haven't uploaded to the SRA, am I out of luck? How do I get those to run? I assume I need to make .bam files based on the github page but how do I get them to run using the software?

The phylogenetics methods section in the manuscript lacks detail. They say the use ML to reconstruct the trees and it looks like the software connects to some tool in EMBL, but what exactly are they using? FastTree, IQTREE? I assume default settings? How is node support determined? Can that be modified? Moreover, why did they mention they can run phylogenetic analyses but not present any trees? Does whatever tree they got actually conform to the current phylogenetic hypotheses for horses?

Same for the clustering analyses, there are several clustering approaches but they don't mention what they were. How can we replicate the work without this information.

In Figure 2, the caption says "The approximate locations where each sample group was sequenced." That wording suggests that they're telling us the location of the lab where the samples were sequenced, but maybe this is a language problem. Shouldn't this refer to collectional localities? The list will need to have more detail. I can read a map but a table with GPS coordinates, cities or at least countries would be helpful. Were they obtained from museums, private collections?

In Figure 5 they have the results of a UMAP analyses, but they don't mention any of that in the methods either. Or is that something that is automatically output by the software?

Where does the SraRunTable.txt come from? No example is provided. Nore is there any information on how to generate it? Thus, I was immediately dead in the water as far as trying this software out.

How important is the directory structure? It seems that the config file would allow me to set up whatever directory structure I'd like and not use the one suggested at on the github page.

In summary, there is much to be done before accepting for publication.

Reviewer #2: Burke et al. developed TEPEAK, a computational method for identifying and characterizing polymorphic transposable elements (TEs) in non-model species without requiring prior annotation or sequence information. By applying TEPEAK to 265 horse genomes, the authors demonstrate its ability to recover both known and novel TE insertions, highlight population structure, and reveal functional impacts on genes across diverse lineages.

1- The authors state that TEPEAK requires a chromosome-level reference assembly. This requirement may limit the tool’s applicability, and I suggest that, if possible, future versions of the pipeline allow genome assemblies at the scaffold level to be used.

2- The manuscript should clarify how the “highest quality” SRA accessions are defined and whether sequencing reads are filtered, trimmed, or otherwise preprocessed prior to analysis.

3- The captions of Figure 1, Table 1, and Figure 2 should be expanded to provide more context and interpretation, ensuring they are fully self-contained.

4- Tool names should be standardized across the manuscript. For example, the text refers to “InSurveyor” in one place and “INSurVeyor” in another. In addition, tool versions should be included where applicable.

Reviewer #3: I have now reviewed the manuscript of Burke et al., entitled “TEPEAK : A novel method for identifying and characterizing polymorphic transposable elements in non-model species populations”. In this work, the authors describe TEPEAK, a novel pipeline to call and analyze transposable elements (TEs) insertion polymorphism in short-read Illumina dataset, and showcase its functionality in a large dataset of horses breeds and related equine species.

Overall, the manuscript is well written and the methods and interpretation are sound. Unfortunately, I wasn’t able to perform a complete test run of the program using other data.

*Positives:

- This is a well integrated pipeline: very convenient to use SRA run table directly, useful downstream analyses.

- The peak calling algorithm, in combination with lookup with DFAM part is innovative and offer a transparent and customizable way to analyze insertion polymorphisms.

- The case study showcases the diverse functionalities of the program and provide interesting insights into modern horses evolution and artificial selection.

*Questions, suggestions:

Introduction:

- “with over 273,000 different families in the most recent DFAM (3.3)” —> the latest DFAM is 3.9 with 4,121,397 total families (26,279 curated), for a total of 2,784 species

- “Class II elements move via a “cut-and-paste” mechanism wherein a transposase enzyme directly mediates the joining of the DNA intermediate to its new site” —> Note that this is not the case ofr RC/Helitrons that are Class II, but use a “peel-and-paste” transposition that effectively copy-and-paste the TE locus.

- “Traditionally, TE families are defined as sequences that share 80% coverage with 80% identity and are thus most often characterized by phylogenetic relationships (although there are families where such characterization is not adhered to)” —> I would argue that biologically speaking, families are a group of homologous sequences expected to originate from a common ancestor locus. The 80-80-80 rule is a useful bioinformatics heuristic but is not grounded in biology.

- “Furthermore, all TEs, to varying degrees, suffer from decay over time which complicates the ability to extract all polymorphic insertions of the same family” —> one could argue that polymorphic families are the most recent one that have colonized a population, and are thus expected to be less degraded than fixed copies.

- “Most current methods require the user to provide a list of one or several TE family consensus sequences to detect polymorphic insertions from that family” —> this is correct, though note that tools like Pantera have been developed to circumvent this problem.

- “leveraging the core idea that elements within the same TE family have similar sizes.” —> I understand the assumption and would think this is correct in most cases, however, I wonder how this could affect an effective detection of LINE-1 elements that are often 5’ truncated and thus do not have a fixed insertion size. (I noticed that some were reported, demonstrating that they can be found, however a discussion on this potential issue would be welcomed)

Methods:

- Please provide additional details on how, in practice, the consensus built are queried against DFAM

- Explain “pseudo-merged population wide VCF”

- Since I wasn’t able to run the program I could not check, but it seems the program does not output VCF, which would be useful for downstream analyses and comparison purposes.

Horses case study:

- “were locked behind a subscription service” —> though this is described later (GIRI Repbase), the issue should be described at this first occurrence in the MS.

- Are exonic insertions predicted to be loss of function? If yes, are they expected to contribute to selected traits? Additionally, one could comment that intronic insertion may be functionally important as some can cause alternative splicing.

*Software testing:

Overall the installation appeared successful following the GitHub instructions. However, I strongly recommend to build a complete docker/apptainer version of the whole software to maximize portability. Conda is not always available on HPC, and virtual environment management is not always easy for non specialists.

- Suggestion: it would be useful to document how to obtain a compatible SRA table via the NCBI website.

- err1 (fixed): “bases” in downloaded SRA run table needed to be changed to “Bases” (uppercase) — the format seem variable.

- err2 (fixed): prefetch (src-toolkit) was not in the environment and had to be installed (conda install sra-tools [via bioconda channel])

- Latest error (could not fix): I first thought it was triggered because I didn’t input SRA for bam, but after trying both (SRA/reads and SRA/bams) I obtained the same error.

Building DAG of jobs...

Using shell: /usr/bin/bash

Provided cores: 16

Rules claiming more threads will be scaled down.

Job stats:

job count

-------

align_species 1

all 1

annotate_genes 1

build_histogram_deletions 1

build_histogram_insertions 1

call_insertions_serial 1

dfam_annotate 1

extract_range 1

get_global_vcf 1

smoove_1 3

smoove_2 1

smoove_3 3

smoove_4 1

smoove_global_vcf 1

write_output_genes 1

total 19

Select jobs to execute...

/home/reviewer/miniconda3/envs/insurveyor-env/lib/python3.9/site-packages/stopit/__init__.py:10: UserWarning: pkg_resources is deprecated as an API. See https://setuptools.pypa.io/en/latest/pkg_resources.html. The pkg_resources package is slated for removal as early as 2025-11-30. Refrain from using this package or pin to Setuptools<81.

import pkg_resources

[Fri Oct 10 10:16:00 2025]

rule align_species:

input: data/albo/albo.fa, data/albo/albo.fa.fai, data/albo/albo.fa.amb, data/albo/albo.fa.ann, data/albo/albo.fa.bwt, data/albo/albo.fa.pac, data/albo/albo.fa.sa, data/albo/albo_samples.txt

output: data/albo/SRR32381021.bam, data/albo/SRR32381019.bam, data/albo/SRR32381018.bam, data/albo/SRR32381021.bam.bai, data/albo/SRR32381019.bam.bai, data/albo/SRR32381018.bam.bai

jobid: 4

reason: Missing output files: data/albo/SRR32381021.bam, data/albo/SRR32381018.bam, data/albo/SRR32381019.bam

threads: 16

resources: tmpdir=/tmp

Running SRA alignment...

2025-10-10T17:16:00 prefetch.3.2.1: 1) Resolving 'SRR32381021'...

2025-10-10T17:16:01 prefetch.3.2.1: Current preference is set to retrieve SRA Normalized Format files with full base quality scores

2025-10-10T17:16:02 prefetch.3.2.1: 1) Downloading 'SRR32381021'...

2025-10-10T17:16:02 prefetch.3.2.1: SRA Normalized Format file is being retrieved

2025-10-10T17:16:02 prefetch.3.2.1: Downloading via HTTPS...

2025-10-10T17:38:15 prefetch.3.2.1: HTTPS download succeed

2025-10-10T17:38:24 prefetch.3.2.1: 'SRR32381021' is valid: 4426778420 bytes were streamed from 4426769921

2025-10-10T17:38:24 prefetch.3.2.1: 1) 'SRR32381021' was downloaded successfully

2025-10-10T17:38:24 prefetch.3.2.1: 1) Resolving 'SRR32381021's dependencies...

2025-10-10T17:38:24 prefetch.3.2.1: 'SRR32381021' has 0 unresolved dependencies

Read 32346618 spots for SRR32381021

Written 32346618 spots for SRR32381021

[Fri Oct 10 10:41:54 2025]

Error in rule align_species:

jobid: 4

input: data/albo/albo.fa, data/albo/albo.fa.fai, data/albo/albo.fa.amb, data/albo/albo.fa.ann, data/albo/albo.fa.bwt, data/albo/albo.fa.pac, data/albo/albo.fa.sa, data/albo/albo_samples.txt

output: data/albo/SRR32381021.bam, data/albo/SRR32381019.bam, data/albo/SRR32381018.bam, data/albo/SRR32381021.bam.bai, data/albo/SRR32381019.bam.bai, data/albo/SRR32381018.bam.bai

shell:

if [ "sra" = "fastq" ]; then

echo "Running custom FASTQ alignment..."

bash -x scripts/align_species.sh -s albo -d data/albo -t 16 -f -l data/albo/albo_samples.txt

else

echo "Running SRA alignment..."

bash src/align_species.sh -s albo -d data/albo -t 16 -f data/albo/albo_samples.txt

fi

(one of the commands exited with non-zero exit code; note that snakemake uses bash strict mode!)

Shutting down, this might take some time.

Exiting because a job execution failed. Look above for error message

Complete log: .snakemake/log/2025-10-10T101559.955648.snakemake.log

Config file:

species: albo

data_dir: data

output_dir: output

zipped_ref_genome_filepath: data/GCF_035046485.1_AalbF5_genomic.fna.zip

zipped_gtf_filepath: data/GCF_035046485.1_AalbF5_genomic.gtf.zip

input_type: sra

sra_run:

filepath: albo_SraRunInfo.csv

number_of_runs: 2

threads: 16

low: 0

high: 10000

gene: y # Include gene annotations

SRA table:

Run,ReleaseDate,LoadDate,spots,Bases,spots_with_mates,avgLength,size_MB,AssemblyName,download_path,Experiment,LibraryName,LibraryStrategy,LibrarySelection,LibrarySource,LibraryLayout,InsertSize,InsertDev,Platform,Model,SRAStudy,BioProject,Study_Pubmed_id,ProjectID,Sample,BioSample,SampleType,TaxID,ScientificName,SampleName,g1k_pop_code,source,g1k_analysis_group,Subject_ID,Sex,Disease,Tumor,Affection_Status,Analyte_Type,Histological_Type,Body_Site,CenterName,Submission,dbgap_study_accession,Consent,RunHash,ReadHash

SRR32628583,2025-07-17 00:24:01,2025-03-10 09:33:13,65729976,19295385812,64307664,293,5183,, https://sra-downloadb.be-md.ncbi.nlm.nih.gov/sos8/sra-pub-zq-820/SRR032/32628/SRR32628583/SRR32628583.lite.1 ,SRX27933450,"=CONCATENATE(A56,""_1"")",WGS,RANDOM,GENOMIC,PAIRED,0,0,ILLUMINA,Illumina NovaSeq 6000,SRP569099,PRJNA1227508,,1227508,SRS24300126,SAMN47286330,simple,7160,Aedes albopictus,Ab21KAHENA104,,,,,,,no,,,,,UNIVERSITY OF FLORIDA,SRA2090678,,public,A9A19922A73CE3C797AF38D2B0D97C71,20C87EE1809F7492FFFDD85E12BA36A7

SRR32628582,2025-07-17 00:23:55,2025-03-10 09:26:26,64654820,19026421115,63104213,294,5087,, https://sra-downloadb.be-md.ncbi.nlm.nih.gov/sos8/sra-pub-zq-820/SRR032/32628/SRR32628582/SRR32628582.lite.1 ,SRX27933451,"=CONCATENATE(A82,""_1"")",WGS,RANDOM,GENOMIC,PAIRED,0,0,ILLUMINA,Illumina NovaSeq 6000,SRP569099,PRJNA1227508,,1227508,SRS24300127,SAMN47286356,simple,7160,Aedes albopictus,Ab21SHIPMN002,,,,,,,no,,,,,UNIVERSITY OF FLORIDA,SRA2090678,,public,103483559021F549A617D88BF6E26236,8B0668B95498D0A4F3A8A9DFC927BC95

**Have the authors made all data and (if applicable) computational code underlying the findings in their manuscript fully available?**

Reviewer #1: **No: ** See comments about installation and implementation in the review.

Reviewer #2: Yes

Reviewer #3: Yes

PLOS authors have the option to publish the peer review history of their article (what does this mean? ). If published, this will include your full peer review and any attached files.

**Do you want your identity to be public for this peer review?** For information about this choice, including consent withdrawal, please see our Privacy Policy .

Reviewer #1: No

Reviewer #2: No

Reviewer #3: No

**Figure resubmission:**

**Reproducibility:**



---

## [Decision Letter · Decision Letter 1]

11 Dec 2025

PCOMPBIOL-D-25-00849R1

TEPEAK: A novel method for identifying and characterizing polymorphic transposable elements in non-model species populations

PLOS Computational Biology

Dear Dr. Burke,

Thank you for submitting your manuscript to PLOS Computational Biology. After careful consideration, we feel that it has merit but does not fully meet PLOS Computational Biology's publication criteria as it currently stands. Therefore, we invite you to submit a revised version of the manuscript that addresses the points raised during the review process.

We look forward to receiving your revised manuscript.

Kind regards,

Adam Ewing

Academic Editor

PLOS Computational Biology

Ferhat Ay

Section Editor

PLOS Computational Biology

**Additional Editor Comments:**

For the most part reviewers were satisfied with the revisions. Reviewer 1 does raise a couple points that I think are reasonable and readily addressable and so I am returning them for your attention. In particular, could you clarify to what extent your method relies on data being present in the SRA?

**Journal Requirements:**

**Reviewers' comments:**

Reviewer's Responses to Questions

**Comments to the Authors:**

Reviewer #1: This is a good attempt at a revision. Many of the comments raised by myself and the other reviewers were addressed but some problems remain.

Those are detailed below:

Lines 64-67 - Cite https://doi.org/10.1038/nrg2165, the first publication of this rule.

L75 - change to "which do encode protiens for autonomous transposition"

L321 - I don't understand how these peaks were determined to be significant. There are plenty of peaks surrounding 7000 bp in the second plot in figure 3. More than the four mentioned. What was the threashold for significance? Is that general region the 'mountain' mentioned in line 194 and the four that were listed in the text are the ones that were considered standouts as described in that section? If so, this should be explained in the figure legend.

I'm still disappointed that there's no evidence that a user who plans to use data not in the SRA will succeed. I would very much like to see that the authors ran such an anslysis. It will greatly enhance the usability for other researchers.

Reviewer #2: The authors have adequately addressed all of my previous concerns, including clarifying TEPEAK’s requirements and detailing data selection and preprocessing, as well as the other points raised in my review. From my perspective, the revised manuscript is suitable for publication.

Reviewer #3: I am satisfied by the authors responses and think that the addition to the manuscript make it fit for publication.

Unfortunately, I wasn't able to perform a successfull run on my own, but I don't think that this involve anything that the authors cannot fix over Github, just I do not have more time to try to fix it right now.

I think the TEPEAK provide significant methodological and scientific developments and will be a useful tool in the TE toolbox.

**Have the authors made all data and (if applicable) computational code underlying the findings in their manuscript fully available?**

Reviewer #1: Yes

Reviewer #2: None

Reviewer #3: Yes

PLOS authors have the option to publish the peer review history of their article (what does this mean? ). If published, this will include your full peer review and any attached files.

**Do you want your identity to be public for this peer review?** For information about this choice, including consent withdrawal, please see our Privacy Policy .

Reviewer #1: No

Reviewer #2: No

Reviewer #3: No

**Figure resubmission:**
---

## [Editor Report · Decision Letter 2]

27 Dec 2025

Dear Mr. Burke,

We are pleased to inform you that your manuscript 'TEPEAK: A novel method for identifying and characterizing polymorphic transposable elements in non-model species populations' has been provisionally accepted for publication in PLOS Computational Biology.

Best regards,

Adam Ewing

Academic Editor

PLOS Computational Biology

Ferhat Ay

Section Editor

PLOS Computational Biology

---

## [Editor Report · Acceptance letter]

PCOMPBIOL-D-25-00849R2

TEPEAK: A novel method for identifying and characterizing polymorphic transposable elements in non-model species populations

Dear Dr Burke,

I am pleased to inform you that your manuscript has been formally accepted for publication in PLOS Computational Biology. Your manuscript is now with our production department and you will be notified of the publication date in due course.

With kind regards,

Judit Kozma
